# Chikungunya Immunopathology as It Presents in Different Organ Systems

**DOI:** 10.3390/v14081786

**Published:** 2022-08-16

**Authors:** Elizabeth M. Traverse, Erin M. Millsapps, Emma C. Underwood, Hannah K. Hopkins, Makenzie Young, Kelli L. Barr

**Affiliations:** Center for Global Health and Infectious Disease Research, University of South Florida, Tampa, FL 33612, USA

**Keywords:** chikungunya, immunopathogenesis, immunology, neurological, cardiac, respiratory, renal, cutaneous, joints, arthritis

## Abstract

Chikungunya virus (CHIKV) is currently an urgent public health problem as high morbidity from the virus leaves populations with negative physical, social, and economic impacts. CHIKV has the potential to affect every organ of an individual, leaving patients with lifelong impairments which negatively affect their quality of life. In this review, we show the importance of CHIKV in research and public health by demonstrating the immunopathology of CHIKV as it presents in different organ systems. Papers used in this review were found on PubMed, using “chikungunya and [relevant organ system]”. There is a significant inflammatory response during CHIKV infection which affects several organ systems, such as the brain, heart, lungs, kidneys, skin, and joints, and the immune response to CHIKV in each organ system is unique. Whilst there is clinical evidence to suggest that serious complications can occur, there is ultimately a lack of understanding of how CHIKV can affect different organ systems. It is important for clinicians to understand the risks to their patients.

## 1. Introduction

The emerging disease, Chikungunya (CHIKV), is currently an urgent problem in public health as high morbidity from the virus leaves populations with negative social and economic impacts [1,2,3]. CHIKV is an arbovirus that is part of the *Togaviridae* family and is transmitted by *Aedes aegypti* mosquitoes in tropical and subtropical regions, and by *Aedes albopictus* mosquitoes in tropical, subtropical, and temperate regions, which are currently the main vector [4,5,6]. While vector transmission of CHIKV is widely known, it is of note that infection can also occur during blood transfusions and vertical transmission during pregnancy, since many different cell types are susceptible to the virus [1]. CHIKV has been reported in 114 countries and in every continent except Antarctica [2,7,8].

Most recently in 2022, as of 5 May, 50,459 cases and seven deaths have been reported in just four months [7]. These cases were mainly reported in Brazil, but have also been confirmed in India, Guatemala, Malaysia and Paraguay [7]. There are currently three genotypes of the virus identified: West African (WA), Asian, and East/Central/South African (ECSA); however, a sublineage has been identified, the Indian Ocean Lineage (IOL), which evolved from the ECSA genotype [1,9,10,11]. The variation in viral genetics for CHIKV has given it a unique epidemiological profile and background that has led to major events which have shaped the history of public health [2,4].

Evidence suggests that Chikungunya-like cases were recorded as early as 1823 in Tanzania, and 1827 and 1828 in the Caribbean, but clinical descriptions also suggest the possibility of outbreaks as far back as the 1600s [1,3,9,12]. However, the emergence of the virus in 1952 in Tanzania led to researchers being able to isolate, identify, and characterize it [1,9,13]. During this time, CHIKV was isolated in the subtropical region and the *Aedes aegypti* mosquito was identified as its mode of transmission into humans [9]. CHIKV remained isolated to Africa and Asia until an outbreak of the virus was identified in 2005 in the La Reunion Islands in the Indian Ocean, which led to the infection of over a third of the population [1]. This outbreak is most remarkable as it is the first evidence of CHIKV being transmitted by *Aedes albopictus* mosquitoes, due to a mutation allowing for enhanced viral infectivity [1,2]. Since *Aedes albopictus* mosquitoes had already successfully occupied much of the globe and could be found in tropical, subtropical, and temperate regions due to high propagule pressure driven by human activities, this mutation allowed for a rapid invasion of the virus, leading to outbreaks in Southeast Asia, North America, South America, and Central America [2,4,14]. The most notable CHIKV outbreaks, since its emergence in *Aedes albopictus* mosquitoes, occurred in Italy in 2007 and in France in 2010, where they raised concerns about the emergence of the virus in temperate regions and how it could potentially affect public health, since these are highly populated and affluent areas of the world [3,15,16].

Even with the lengthy epidemiological history of CHIKV, data suggest that outbreaks may have been missed due to limitations in diagnosis, surveillance, and non-symptomatic infections [9,17]. Diagnosis of CHIKV can be challenging as it can be easily misdiagnosed, so a serological test is needed in order to confirm the presence of the virus [2,5]. CHIKV is characterized by acute fever and joint pain that can lead to chronic arthritis which can be severe and debilitating for months to years [5,6,18,19]. Acute infection and symptoms can last 3–10 days and are treated with drinking plenty of fluids, rest, and the appropriate antipyretics or analgesics, as prescribed by a physician for pain [5,6]. Rare complications include uveitis, retinitis, myocarditis, hepatitis, nephritis, bullous skin lesions, hemorrhage, meningoencephalitis, myelitis, Guillain-Barré syndrome, cranial nerve palsies, or the relapse of rheumatologic symptoms, among others [18,20]. Death from CHIKV is rare and usually only occurs in elderly individuals with comorbidities, or in children [18,20]. However, approximately 40% of patients who are positive for CHIKV suffer with long-term debilitating effects and a compromised quality of life due to chronic pain and arthritis, affecting their mobility and requiring life-long treatment [1,6,18,19]. Unfortunately, there is no current specific treatment or vaccine for the virus, but several vaccines are currently in various stages of clinical development [5,9,21,22,23,24,25].

CHIKV has the potential to affect every organ of an individual, leaving patients with lifelong impairments which negatively affect their quality of life [1,18,20]. There is a high demand for treatment and vaccine research for CHIKV, since the high rate of morbidity caused by the virus creates significant social and economic impacts [1,2]. This review will show the importance of CHIKV in research and as a public health issue, by discussing the immunopathology of the virus as it presents in different organ systems, affecting the health and well-being of individuals. Sources used in this review were found on PubMed, using “chikungunya and [relevant organ system]” as broad search terms.

## 2. General Chikungunya Infection of the Human Host

The general immunology of CHIKV has been summarized in a review by Petitdemange et al. [26]. Briefly, CHIKV has a broad tropism, which allows it to enter and replicate within a variety of human cells, including endothelial, epithelial, and fibroblast cells among others [27,28]. The known cellular attachment and entry factors include prohibitin (PHB), a protein with many functions which is ubiquitously expressed, matrix remodeling associated 8 (MXRA8), a membrane protein with connections to signaling pathways, basigin (also known as BSG or CD147), a plasma membrane protein, T cell immunoglobin and mucin 1 (TIM-1), which is expressed in a variety of cell types and is a receptor for several viruses, and DC-specific intercellular adhesion molecule-3-grabbing non-integrin (DC-SIGN), which has been shown to enhance alphavirus infections [29,30,31,32,33,34,35].

The full virion replication cycle is typically complete in 8 h [28]. Once released by the infected cell, CHIKV can circulate throughout the body using the lymphatic system and the bloodstream, leading to several sites of infection [36]. Being a single-stranded RNA virus, toll-like receptors (TLR) 7 and TLR8 can detect the viral genome, and as the CHIKV viral genome replicates, it has a double stranded RNA intermediate which may allow detection by TLR3 [36,37]. When TLR7 is activated, it begins a cascade which results in the production of interferon (IFN)-β and later IFN-α, as well as the stimulation of B cells [38]. There is also a marked type I IFN response in clinical cases, as IFN-α is correlated with viral load [36]. This innate immune response recruits inflammatory cells to sites of infection in an effort to control viral spread, possibly leading to disease symptoms as host tissues are invaded and cells die [36]. In infected individuals, there are elevated plasma levels of interleukin (IL)-5, IL-6, IL-7, IL-8, IL-10, IL-12, IL-15, and IFN-α [39,40]. These proinflammatory interleukins suggest that there is a strong inflammatory response during the acute phase of the disease, followed by the release of proinflammatory migration inhibitory factor (MIF) from activated macrophages which recruit leucocytes to the infected areas [39].

Following the release of proinflammatory cytokines and chemokines, adaptive immunity is activated [39]. CD8 T lymphocytes are activated and peak on the first day, staying elevated for 7–10 weeks in those experiencing arthritis symptoms [40]. Anti-CHIKV antibodies for the disease have been shown to be effective; in mice without B and T cells, anti-CHIKV monoclonal antibody administration alone was enough to prevent disease [41]. The humoral immune response is triggered by the glycoprotein E2 on the virus’s envelope, with a majority against the epitope E2EP3 in the N-terminus of the protein [41]. It is uncertain how long these antibodies stay in the body and how long lasting acquired immunity of the disease is [42].

Overall, there is a significant inflammatory response during CHIKV infection, which may result in the clinical pathogenesis observed in diseased patients affecting several organ systems, such as the brain, heart, lungs, kidneys, skin, and joints [36]. Organ-specific immunopathogenesis is discussed below, which is important for clinicians to understand due to the numerous severe presentations of CHIKV infection, which may be misdiagnosed or missed entirely (Figure 1).

## 3. Clinically Disturbing Neurological Complications

Chikungunya causes numerous neurological syndromes and disorders, including encephalitis, meningoencephalitis, optic neuritis, Guillain-Barré syndrome, paralysis, acute disseminated encephalomyelitis, and neonatal hypotonia [43,44,45,46,47]. Of the neurological complications of CHIKV, encephalitis, which is inflammation of the brain parenchyma, is the most common and can result in encephalopathy [43,47,48,49,50,51]. Long-term consequences of encephalitis include cognition issues, mood changes, depression, confusion, and memory loss [52].

The immunology of a CHIKV infection in the brain and how complications such as encephalitis arise is understudied, but research is ongoing, and many studies have provided insight. In a zebrafish model, a real-time CHIKV infection was observed, revealing that the virus infected the brain parenchyma and persisted longer than in other tissues [53]. CHIKV was detected in neurons and glial cells [53]. This model showed a protective type I IFN response, produced primarily by neutrophils, and seemed to control CHIKV replication levels and pathogenesis [53]. In order to access the brain, CHIKV seems to first infect endothelial cells of the brain vasculature, which would more easily lead to infection of the brain parenchyma, potentially leading to encephalitis [54].

Consistent with the zebrafish model, cytokine profiling of individuals with neurological disease during CHIKV infection showed an increase in TNF-α, IFN-α, and IL-6 in CSF samples, characteristic of a type I IFN response [47,55,56]. This response acts as a positive feedback loop to create more type I INFs, activating the JAK/STAT pathway [56]. The cytokine profiling also revealed increased levels of IFN-γ, a type II IFN which is traditionally seen in an antiviral response [47,55,56]. Other animal models which involve CHIKV infection of the brain reveal more information. TLR3 was upregulated in the brains of CHIKV-infected mice [47,57]. When treated with a TLR3 agonist, an upregulation of IFN-β, among other proinflammatory cytokines, was shown to be protective and promoted viral clearance from the brain [57]. A model using *Cynomolgus* macaques showed activation of astrocytes, the most common cell type in the central nervous system [37,58]. Here, astrocytes had an increased TLR2 expression, which could have been activated by damaged neurons that release DAMP molecules which are then detected by TLR2 on glial cells [37]. In a glioblastoma model, infection with CHIKV resulted in apoptosis and an increased expression of IL-1, TNF-α, IL-6, and CXCL9 [59]. While this model is inherently lacking in a type I IFN response, the increase in these proinflammatory cytokines does indicate activation of the innate immune response [59].

Other important immune cells, in addition to astrocytes, are microglia, which are the macrophages of the brain [60]. In CHIKV-infected macaques, it was shown that CHIKV replicates in lymph tissue and persists in activated macrophages for months [61]. In a microglia-specific study, researchers infected CHME-3 (human microglial clone 3) cells with CHIKV [62]. These cells showed no cytopathic effects when infected, and an MTT assay confirmed cell viability 72 h postinfection [62]. Studies with transmission election microscopy showed mostly morphologically normal CHME-3 cells with vacuolation and abnormally enlarged mitochondria [62]. The researchers could also observe the presence of replicating virions in the CHME-3 cells [62]. Interestingly, in another model of CHIKV-infected microglial cells, different protein expression levels were seen, including a down regulation of proteins involved in the JAK/STAT pathway [63]. Since this observation was connected to the nsP2 and capsid proteins of CHIKV, it is possible that these proteins are connected to viral evasion of the host immune system [63]. Much more research needs to be conducted to elicit a full view of how CHIKV can invade the different cell types of the brain, and the resulting damage that can ensue.

## 4. Increasingly Common Cardiovascular Presentations

There are many reports that detail a wide range of cardiac symptoms involved during CHIKV infection, including arrythmias, palpitations, abnormal electrocardiograms and echocardiograms, myocardial infarctions, heart failure, cardiac arrest, cardiomegaly, and myocarditis among others [64,65]. While some of these presentations may be due to the body’s reaction to the infection, including the innate immune response and inflammation, other symptoms result from direct viral invasion of cardiac tissues [64,66,67]. In the heart, viral infection is detected by toll-like receptors, of which TLR3 and TLR7 are of importance, leading to a proinflammatory, type I interferon response [68]. Upon activation of the innate immune response, natural killer cells (NKCs) begin to target infected cells, destroying them to limit infection [69]. In addition to these NKCs, cytokines such as IL-1, IL-2, TNF-α, and IFN-γ are produced, some with more cardioprotective effects than others [69]. CD8 T lymphocytes then begin to enter the cardiac tissues and limit infection, primarily by destroying infected cells, the debris from which can then further stimulate cell mediated cytotoxicity by acting as antigens for additional T cells [69].

One interesting consequence of CHIKV infection which is found across nearly all age groups is myocarditis, which results when the virus directly infects the cardiac tissues and causes damage [64,67,70]. Known receptors for CHIKV exist on the surface of cardiomyocytes, including PHB, MXRA8, and CD147 [31,33,71,72,73]. Additionally, it has been shown that CHIKV viral antigen is found in heart tissues, and a higher CHIKV viral titer has been linked to increased cardiac damage [74,75,76,77]. The previously described immune pathway begins upon CHIKV infection, however, in some cases, the fragile balance that is required to keep the immune system from harming the host is askew [67,69]. When this occurs, NKC-like cells which produce perforin, TNF-α, and an overactivation of T cells, are deleterious to cardiac tissues and not always seen at the clinical level, making diagnosis and intervention difficult [67,69]. This kind of rapid damage is repaired by fibrotic tissues, which are not contractile heart muscle cells and could possibly lead to other cardiac conditions such as dilated cardiomyopathy [69,70,78,79].

Interestingly, overactivation of TLR3 is more closely associated with pathogenesis of myocarditis [68]. Furthermore, TLR3 is upregulated in patients infected with CHIKV, while TLR3-knockout mice models are more susceptible to CHIKV, have increased pathogenesis and have higher viral titers [80]. These knockout mice show a correlation between the loss of TLR3 and a lack of CHIKV IgG, which would contribute to a neutralizing response [80]. However, in the CHIKV replication cycle within mammalian cells, the double-stranded intermediate of CHIKV is protected from recognition by the pattern recognition factors of spherules, or viral replication compartments [36]. Additional research needs to be conducted to better understand the mechanism by which CHIKV infection elicits cardiac damage and myocarditis, given this apparent contradiction concerning TLR3 activation.

## 5. Rare Respiratory Complications

While rare, CHIKV seems to be capable of infecting the lungs, causing acute respiratory distress syndrome (ARDS) and some cases of respiratory failure, which have resulted in the death of at least one patient [51,81,82,83,84,85,86,87,88]. In addition to human patients, mouse models of CHIKV infection have also exhibited inflammatory cells in the alveolar sacs at the end of bronchioles in the lungs [74]. ARDS is a type of inflammatory lung injury that leads to increased vascular permeability of the lungs, increased lung weight, and loss of tissue that is aerated, and has mild, moderate, and severe severity categories [89]. From a clinical perspective, ARDS presents as hypoxia, altered chest radiography, and diffuse alveolar damage, sometimes requiring the use of a mechanical ventilator, and with a mortality rate as high as 46% worldwide [90]. While ARDS can be seen due to injury from an infecting virus, CHIKV is an unusual culprit [88,89,90]. Similar to other organ systems, viral infection of the lungs elicits a type I IFN response accompanied by TNF-α, IL-1, IL-6, IL-8, and IL-12 [91]. Viral respiratory infections also manifest primarily in a Th1 response, where IFN-γ, CD8 T cells, and NKCs all play a role in viral clearance, a mechanism seen for CHIKV in other organs [36,91]. These innate immune mechanisms may result in the inflammation seen in ARDS patients being an inflammatory condition [89].

When it comes to clinical presentations and differential diagnosis, the absence of a respiratory illness or respiratory symptoms is an indication of an arboviral disease, such as CHIKV [92]. Therefore, clinicians may misdiagnose some CHIKV patients due to their abnormal presentations [88]. However, what is recognized is how respiratory symptoms due to concomitant infections with CHIKV and another illness can be a risk factor for severe CHIKV and/or hospitalizations [50,93]. Additionally, pre-existing conditions which involve the lungs may also be a risk factor [50]. Concurrent viral infections can alter how the body responds to an illness, either with attenuation of the inflammatory response, or the enhancement of it, where attenuation might allow an infection to persist, and enhancement may increase cytopathic effects [94,95,96]. While rare, respiratory illness such as ARDS can occur during CHIKV infection, though the existence of pre-existing or concomitant respiratory illness presents a more concerning clinical challenge during infection, leading to the necessity for additional data on how CHIKV interacts with the lungs.

## 6. Potential Renal Complications and Concerns for Organ Transplantation

There is not much data which reveal how CHIKV infection affects the kidneys, but what little clinical evidence exists shows that CHIKV can cause acute kidney injury (AKI), usually if rhabdomyolysis, acute interstitial nephritis, thrombotic microangiopathy, or a renal lesion is already present [97,98,99]. Furthermore, kidney disease, as a pre-existing condition, can be exacerbated, resulting in worse symptoms of infection, hospitalization, or death [99,100,101,102]. Unfortunately, due to the lack of data, not many conclusions can be made about the immunological response to CHIKV infection in the renal system.

The kidneys are one of the most common organs to be transplanted, which presents an interesting concern for CHIKV infection [103]. In addition to acquiring CHIKV via a mosquito bite after transplantation, it is possible for an organ recipient to become infected from a CHIKV-positive donor [104,105]. Transplantation can be deferred until the donor has cleared the infection [105]. When a patient undergoes an organ transplant, they are placed on a regimen of immunosuppressive medications which alter the body’s response to viral infections [97,106]. Immunosuppressive medications decrease the expression of genes that encode for proinflammatory cytokines, including the IFNs and TNF-α, while other medications inhibit lymphocyte development and reduce IL-1, IL-2, IL-3, IL-4, IL-5, IL-6, IL-10, and IFN-γ [97]. A number of these cytokines are required for pathogenesis and the inflammatory response to CHIKV; thus, suppression seems to reduce the symptoms and severity of illness, including prolonged arthralgia [97,104,107]. However, this also presents a question as to whether or not viral clearance has occurred, requiring more study [108].

## 7. Cutaneous and Mucocutaneous Presentations

There are several different types of cutaneous or mucocutaneous presentations of CHIKV, reported in all age groups and of varying severity [109,110,111,112,113,114,115,116,117,118,119,120,121,122]. As high as 50% of CHIKV patients will have some sort of mucocutaneous presentation, most commonly a maculopapular rash, which shows up 3–5 days post fever onset and alleviates 3–4 days later [121]. Additional presentations include general skin lesions, panniculitis lesions, hyper- and hypopigmentation, vasculitic lesions, oral ulcers, genital ulcers, toxic epidermal necrolysis, and nasal necrosis, among others [81,87,109,110,111,112,113,114,115,116,117,118,119,120,121,122,123]. Among these presentations is a pigmentation of the nose, also known as “Chik Sign” [111]. Most of these conditions are asymptomatic, however, a few can cause pain to the patient [109]. Additionally, it appears that CHIKV infection can also cause the exacerbation of pre-existing dermatoses [115,121].

While many case reports only describe the symptoms involved, some also give results of Tzanch smears, a diagnostic tool to corroborate cutaneous diagnoses, which showed lymphocytes, acantholytic cells, necrotic keratinocytes, and a small number of neutrophils in bullae [124,125]. Additionally, immunohistochemistry from these patients showed an infiltration of CD8 T cells [124]. These findings align themselves with the proinflammatory response exhibited by other organ systems, as well as showing how the skin is infiltrated by a variety of immune effector cells [36,38]. Experimentally, skin fibroblasts infected with CHIKV demonstrated an upregulation of MX1, IFIT1, IFIT3, and ISG15, which are all stimulated by interferon, showing the type I IFN response seen in other organ systems [126]. The IFIT proteins seemed to have been involved in the inhibition of CHIKV replication in these cells [126]. In the same study, DDX58, STAT1, OAS3, EIF2AK2, and SAMDD1 were also upregulated, all of which are defense response proteins [126]. Along with the clinical data, these findings support the theory that the innate immune system launches a type I IFN response to attack and clear CHIKV in skin.

CHIKV is transmitted through mosquito bites, and after going through the dermis and epidermis layers of the skin, the virus then attaches to fibroblasts, starting the replication process [127]. When CHIKV enters the skin of a human, it includes saliva from the bite of a mosquito, which may alter how the body responds to the virus, as was examined in a study by Agarwal et al. [128]. This study found that in mice, there was an infiltration of inflammatory cells at the site of the mosquito bite, where hemorrhage was present in the dermis of CHIKV-positive bites, but not negative bites [128]. The cytokines IL-4 and IL-10 were upregulated in the skin while TLR3, IL-2, IFN-γ, TNF-α, and IFN-β were all downregulated, implying that the presence of mosquito saliva actually aids CHIKV by down regulating the proinflammatory mechanisms required to clear CHIKV [36,39,40,128]. IL-4 and IL-10 are parts of the Th2 immune response, which directly counteracts the Th1 response, furthering the immunosuppression, possibly allowing CHIKV to take hold in the host [128,129,130]. These are interesting findings which should be further explored due to their implications during human infection by mosquitos.

## 8. Joint Inflammation and Arthritis, the Most Common Presentation

From a clinical perspective, arthralgia is a common symptom of CHIKV infection, with approximately 87% of all cases having this presentation as part of the acute stage of infection [131]. This arthralgia can persist or relapse into a chronic condition, known as chronic chikungunya arthritis syndrome, which can last for weeks, months, or years, affecting as many as 40.2% of patients [131,132]. Chikungunya arthritis symptoms can mimic those of rheumatoid arthritis (RA), so much so that in some cases the symptoms are similar enough that they qualify as RA, according to the diagnostic criteria set by the American College of Rheumatology [52,132,133].

There are other alphaviruses which cause persistent joint pain, including Ross River virus (RRV), Barmah Forest virus (BFV), Mayaro virus (MAYV), and o’nyoung’nyoung virus (ONNV), which CHIKV is closely genetically related to [134,135]. In RRV, joint pain is associated with increased levels of IL-17 produced by T cells that circulate in host tissues to fend off the virus [134]. IL-17 can promote inflammatory cytokines such as TNF-α and IL-1 [134]. In mice, it has been shown that both CD4 and CD8 T cells remain within joint tissues even after RRV viral clearance [134]. This is similar to clinical CHIKV data, as humans and animals infected with CHIKV show an increased Th17 inflammatory response, marked by an increase in IL-6, IL-17, and IFN-α [133,136,137,138]. The generation of Th17 cells by naïve T cell differentiation requires TGF-β, IL-1, IL-6, or IL-21 [139,140].

The Th17 response is implicated in both autoimmune diseases as well as in the removal of extracellular bacteria, and is marked by a subset of CD4 T cells which produce IL-17, IL-6, and TNF, and seem to require IL-23 for pathogenicity [139,140]. In addition to the proinflammatory cytokines produced by these cells, receptors on the T cell surface encourage trafficking into diseased tissues [140]. A Th17 response is seen in animal models of RA, which chikungunya arthritis can mimic [133]. In RA patients, there is an increased level of IL-17 in joints, where a higher level of IL-17 is associated with additional pain, revealing the pathogenic nature of the cytokine [140]. In both RA and chronic chikungunya arthritis, the IL-17 response may drive bone destruction and increased production of inflammatory cytokines and chemokines [133]. Methotrexate (MTX), a common treatment for RA, has been shown to reduce the symptoms of chikungunya arthritis, possibly due to the drug’s ability to reduce antigen-dependent T cell proliferation [108,133,141]. Specifically, MTX has been shown to reduce the production of IL-4, IL-6, IL-13, and TNF-α, some of which are important factors for Th17 cell differentiation [139,140,141]. One concern of the use of MTX in patients with viral infection is a possible reduction in the immune response and a delayed viral clearance, however, in vitro studies have shown that MTX did not alter the antiviral response to CHIKV infection [141,142]. Additional research should be conducted to ascertain the full effectiveness of MTX as a treatment for chronic chikungunya arthritis. A few other medications, including hydroxychloroquine, meloxicam, Ribavirin, and immunosuppressants, have also been used to treat CHIKV arthritis with varying success, as is described in the review by Sales et al. [21].

Outside of a potential Th17 response, it is known that chronic chikungunya arthritis syndrome is also accompanied by increased levels of CD4 T cells, IL-6, IL-12, and macrophage inflammatory protein [143]. However, this is contrasted in the acute phase of infection, as an initial CD8 response moves to control viral replication of CHIKV in infected tissues, exhibiting an IFN-1 response with IL-4, IL-6, IL-7, CCL2, CCL4, CXCL10, and macrophage migration inhibition factor [138,143]. These CD8 T cells and later-recruited NKCs work to remove the virus within 7–10 days post infection [143]. CD4 T cells also begin the neutralizing antibody response [138].

In mouse models, joint pain is associated with the increased level of CD4 T cells and viral clearance, rather than viral titer [138]. Teo et al. showed that joint pain was not associated with macrophage or neutrophil migration to joint tissues, while Her et al. showed the opposite in TLR3-knockout mice; severe joint inflammation was the consequence of infiltration by myeloid cells, including neutrophils and macrophages [80,138]. In this study, the TLR3-knockout mice also showed a reduction in CD4 T cells, which are necessary for pathology, and possibly represent a relationship between TLR3 and the acquired immune response [80]. In order to fully elucidate these discrepancies, additional research needs to be conducted.

## 9. Conclusions

In the past several decades, chikungunya has become an ever-growing concern due to increasing cases, abnormal and deadly presentations of disease, and a lack of treatment or vaccine [3,144]. CHIKV has the potential to affect nearly every organ of an individual with its broad tropism, however, a distinct Th1 type I IFN response is the most common immunological defense employed to control and eliminate the infection, with a Th17 response in chronic joint manifestations, and an initial Th2 response in the cutaneous system [2,128,139,140]. Though rare, there are cases in which individuals die due to CHIKV infection, primarily the elderly, the very young, or those who are immunocompromised [18,20,145]. In addition, there has been a global expansion of the viral vector, which is furthered by climate change, and confers an additional disease risk to more countries and populations [3,119,146]. Clinical and preclinical vaccines are in development, detailed by de Lime Cavalcanti et al., but more research needs to be conducted to fully ascertain the clinical pathology of chikungunya infection in an effort to protect global populations [22,147].

## Figures and Tables

**Figure 1 viruses-14-01786-f001:**
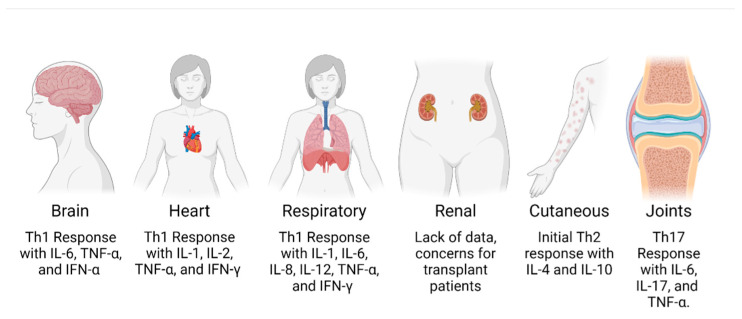
Generalized schematic of the major organ systems affected by CHIKV infection and their associated T cell and cytokine response. Figure generated in BioRender.

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
