# Peer review of "Chikungunya Immunopathology as It Presents in Different Organ Systems"

_viruses, 2022, doi:10.3390/v14081786_

Round 1

Reviewer 1 Report

The scope of this review article is very interesting, describing the different immunopathological manifestations in different organs caused by CHIKV infection. I have a few comments on the article:

-Section 6 appears to be split into renal complications and organ transplantation. I do not completely understand why these two aspects were put together in the same paragraph. In addition, the part about organ transplantation mostly describes the effect of immunosuppressive drugs on the manifestation of the illness (even improving it) and therefore I am not sure whether this part fits into the scope of the article (immunopathology).

-It would be nice to expand section 8 by putting more focus on the clinical/immunopathological differences between acute CHIKV arthritis and chronic CHIKV arthritis.

-The conclusion is not very strong. Lines 368 to 374 are too general and seem to be superfluous to the scope of the article and do not add much. There is also too much emphasis on vaccine. Please consider rewriting the conclusion.  

-Linguistic comments (consider making the following changes, indicated in bold):

Line 37: These cases were mainly reported in Brazil...

Line 49-50: CHIKV remained isolated to Africa and Asia until an outbreak of the virus was identified in 2005...

Line 51: ....in the Indian Ocean that led to infection of over a third of the population.

Line 58: The most notable CHIKV outbreaks since its detection in Aedes albopictus mosquitoes occurred in Italy in 2007 and in France in 2010, as they raised concerns about emergence of the virus in temperate regions....

Line 69: ...and the appropriate antipyretics or analgesics prescribed by a physician for pain.

Line 73-74: ...only occurs in elderly individuals with comorbiditiesor in children.

Line 74: "However, 40% of the victims of this virus will suffer from long-term debilitating effects...."

Also, please be more specific. I think the word "victim" is too colloquial and general. Is it 40% of persons testing positive for CHIKV, or those being diagnosed with arthritis during the acute phase? 

Line 75: Please put a comma behind "arthritis".

Line 83: ...the importance of CHIKV in research and as a public health issue by focusing on the immunopathology of the virus....

Line 84: ....as it presents in different organ systems, affecting the health....

Line 85: Consider deleting "...populations, and the globe".

Line 87: ...broad search terms.

Line 97: ...which may allow detection by TL3.

Line 103: Write out the term "interleukin" if not already done so earlier in the document.

Line 104: Consider deleting "it suggests".

Line 105: ...inflammatory response during the acute phase of the disease...

Line 106: ...from activated macrophases that recruit leucocytes....

Line 110-111: ...have been shown to be effective....

Also, "Antibodies for the disease" is too unspecific. Do the authors mean anti-CHIKV antibodies? Also name the mouse model that lacks B and T cells. 

Line 148: Other animal models that involve CHIKV infection of the brain...

Line 151: ...to be protective and promoted viral clearance....

Line 160: Another important immune cell in addition to the astrocyte is the microglial cell....

Line 163: Consider deleting "cells" behind "human microglial clone 3" as it is indicated twice.

Line 183: The words "activation of" does not seem to fit into the sentence. Consider changing the phrasing or deleting these two words.

Line 183-184: Consider deleting "CHIKV is a positive, single-stranded RNA virus, which would be detected by TLR7" as this has already been mentioned before in the document.

Line 189-190: ....primarily by destroying infected cells, from which the debris can then further stimulate...

Line 196: Consider deleting "has".

Line 197: ...signaling pathways and that exhibits low expression...

Line 200: ...in heart tissues, and higher CHIKV viral titer....

Line 204: ...overactivation of T-cells are deleterious...

Line 205: "cardiac tissue"

Line 238: Therefore, clinicians may misdiagnose some CHIKV patients...

Line 249: Transplantation

Line 277: ...epidermal necrolysis, and nasal necrosis....

Line 279: Consider putting a comma behind "however".

Line 287: ...as well as showing....

Line 312: ...87% of all cases having this presentation as part of the acute stage of the infection.

Line 329: ...as well as in the removal of....

Line 353: ...and viral clearance, rather than viral titer.

Line 355: ...Her et all showed the opposite in TLR3 knockout mice...

Line 362: Consider deleting "a mosquito vectored emerging disease"

Line 364: ...and absence of vaccine....

Author Response

The following has been changed due to your review:

Section 6 has additional detail to explain the inclusion of transplantation with renal complications (please see line 260). As immunosuppressive medications by nature alter potential immunopathology, the authors feel this is a necessary addition to the manuscript. 

Section 8 has been expanded upon to include more detail on the acute condition (please see lines 356 to 363).

Conclusion was reworded/rewritten.

For the specific grammatical alterations, several were adopted, other lines were deleted or changed as a whole, or the suggestions were deemed as stylistic differences that do not change the understanding of the manuscript. 

We thank you for your commentary.

Reviewer 2 Report

The review by Kelli L. Barr and colleagues provided a thorough review on Chikungunya immunopathology in different organ systems, from brain to joints. It is important to understand the clinical pathology of CHIKV infection in different organs for developing vaccine and therapies. One comment is the broad tropism for CHIKV is probably due to the broad distribution of CHIKV receptors/entry factors. It will be helpful to include literatures on this. 

Author Response

The following changes to the manuscript have been made due to your commentary: 

Additional literature and details have been included regarding Chikungunya Virus attachment and entry molecules (please see lines 93-99). 

We thank you for your commentary.